# World Model Anomaly Detection with a Latent Linear Prior

## Abstract

Model-based reinforcement learning (MBRL) learns world models—internal simulators of environment dynamics—to plan by imagining future trajectories. However, when these models incorrectly predict state transitions, they generate unrealistic states that mislead agents into learning delusional policies. Inspired by human vision, we propose anomaly detection in world model with **L**inear **P**rior (LP), a three-stage approach that 1) enforces a lightweight linear prior on successive latent states, 2) flags generated states that deviate from this prior, and 3) removes their contribution during agent learning. On the challenging Atari100k benchmark, LP-assisted GRU and Transformer based MBRL agents achieve competitive results while requiring less value updates with minimal additional computational cost. Notably, by suppressing false value updates with LP, DreamerV3 boosts human-normalized mean score by 9% while requiring less than 90% of the value updates. We release our implementation at https://anonymous.4open.science/r/lp-dreamer.

## 1 Introduction

Deep reinforcement learning (RL) has achieved remarkable success recently, including in arcade games (Mnih et al., 2015; Schrittwieser et al., 2020; Hafner et al., 2021; 2025), real-time strategy games (Vinyals et al., 2019; OpenAI, 2018), board games (Silver et al., 2016; 2018; Schrittwieser et al., 2020), and games with imperfect information (Schmid et al., 2021). Despite the successes, sample inefficiency remains a problem – limiting the agents' ability to learn from limited interactions. Some model-based reinforcement learning (MBRL) methods attempt to tackle the sample inefficiency by approximating agents' real interactions with the environment with a learned *world model* — an internal simulator that predicts environment dynamics, rewards, and continuation signals — to reduce expensive real–world interactions (Sutton & Barto, 2018; Sutton, 1991; Ha & Schmidhuber, 2018b). These models can take the form of a sequence modeler such as transformers (Zhang et al., 2023; Micheli et al., 2022; Chen et al., 2022), GRUs (Hafner et al., 2019; 2021; 2025), or hybrid state-space architectures like MAMBA (Wang et al., 2024), to generate latent rollouts (Nair et al., 2018; Nasiriany et al., 2019; Hafner et al., 2025) for planning and policy updates.

However, learned world models contain imperfections that can lead to unwanted generalizations (Xu et al., 2025; Zhang et al., 2024; Xing et al., 2024; Jesson et al., 2024; Aithal et al., 2024), resulting in inaccurate value updates and misleading downstream policy optimization (Buckman et al., 2018; Janner et al., 2019). For simplicity, in this study, we refer to unrealistic predictions of environmental dynamics by world models as *anomalies* and seek to exclude them from agent learning. To see why this is important, take, for example, the world model shown in Figure 1. An agent trained on its rollout could mislead itself getting to a state that is unrealistic in the real environment which would lead to delusional value updates, such as believing there is an *extra* ice floe to cross the river in *Frostbite* or an *extra* fighter to defeat in *Kung Fu Master*.

In MBRL, such world model anomaly can result in a policy based on delusional values or unsafe behaviors (Janner et al., 2019; Bengio et al., 2024; Zhao et al., 2025). With that in mind, several questions interest us: *can we detect these states? can we suppress their negative impact?*

We draw inspiration from the human visual cortex that understands *naturalness* by mapping natural scenes onto straighter, low-curvature trajectories in representation space (Hénaff et al., 2019)—making real inputs *straightforward* to interpret—while artificial scenes induce higher curvature. This phenomenon—known as

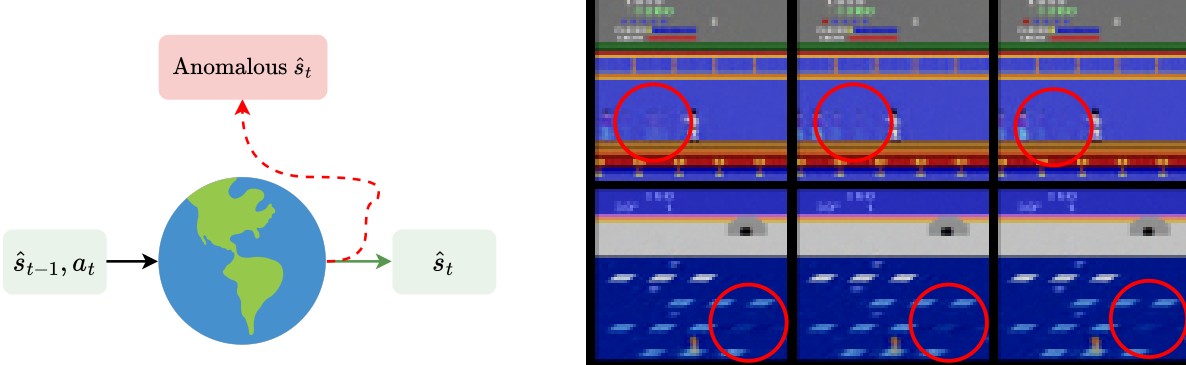

Figure 1: *Left:* World model can *make mistakes.* incorrectly predicted state transitions produce false targets, which results in a subpar policy. *Right:* Example of DreamerV3 world model anomalies in the reconstructed pixel space in games *Kung Fu Master* and *Frostbite* detected our Linear Prior(LP) method.[1]

*perceptual straightening*—suggests that plausible sensory inputs tend to follow simpler, more linear paths in the latent space. Inspired by this, we ask:

*Can we filter anomalous world model states by enforcing a linear prior over successive latent representations?*

Concretely, this requires constraining latent dynamics to evolve along approximately straight paths in representation space and suppressing value updates for state transitions that deviate from this prior, effectively treating high-curvature transitions as anomalies.

In this work, we propose world model state anomaly detection with *Linear Prior (LP)*, which (i) augments world model with a linearity-consistency loss, (ii) flags anomalous rollouts using a linear prior, and (iii) removes delusional value updates during policy learning. First, we define and implement a consistency loss that encourages hidden state dynamics to remain nearly linear (eq. (1)). Next, we derive a provable anomaly detection criterion based on train-time linearity metrics (§3.4). Then, we show how to remove the influence of rejected anomalous states for agent learning (§3.5). Finally, we implement LP on both GRU-based DreamerV3 and transformer-based STORM (§4). For LP-DreamerV3, we show that canceling value updates on anomalous states yield significant performance gains. In fact, with less than 90% value updates of DreamerV3, LP-DreamerV3 yields 9% better human normalized mean than DreamerV3 on the Atari100k benchmark (Bellemare et al., 2013; Kaiser et al., 2020). Interestingly, we show that transformer-based world models such as STORM naturally exhibit high-linearity hidden representations, resulting in far fewer rejections with LP-STORM compared to that of GRU-based approach. Beyond the immediate gains in Atari performance, LP points toward a broader paradigm for trustworthy, sample-efficient decision making in any domain where learned simulators stand in for reality.

## 2 Related work

**World models in MBRL**

MBRL learns world model of its environment through online or offline data, and use it for better generalization, sample-efficiency, and decision-making. One of the first ideas of using a world model for MBRL can be attributed to Dyna (Sutton, 1991) that learns a one-step predictive world model, which the agent uses to perform policy updates in the imagination. Following the success of deep neural networks, there have been many recent works. Few initial works focused on modeling the lower dimensional proprioceptive environments (Silver et al., 2017; Henaff et al., 2017; Wang et al., 2019; Wang & Ba, 2020). However, Ha &

---

[1]Stochastic mapping from state to observation in modern world models make a direct qualitative evaluation of states from predicted observations challenging. For a detailed discussion, we guide the readers to §H.

Schmidhuber (2018a)'s work shifted the focus more on building world models that operate on the pixel space. Ha & Schmidhuber (2018a) trains a world model from trajectories of pixel observations and actions, learning a vision network that learns abstraction of the image through a variational autoencoder(VAE) (Kingma & Welling, 2013) and a memory network that uses RNN to predict next state given the current state and action.

Following the success of Ha & Schmidhuber (2018a), SimPLE (Kaiser et al., 2019) proposed one of the earliest world model applied to Atari100k benchmark. The world model learns to predict next observation and environment rewards, given previous observation and action, essentially acting as a walk-in replacement for the real environment, which a Proximal Policy Optimization (PPO) agent (Schulman et al., 2017) uses to improve its policy. PlaNet (Hafner et al., 2019) improves world modeling by introducing Recurrent State-Space Model (RSSM) for learning dynamics in the latent space using a Gated Recurrent Unit (GRU) (Cho et al., 2014). It also uses model predictive control (MPC) for planning using the learnt world model, becoming one of earliest works to successfully use this on the DeepMind Visual Control (DMC) tasks (Tassa et al., 2018).

With the success of Hafner et al. (2019) came Dreamer (Hafner et al., 2020; 2021; 2025). Dreamer (Hafner et al., 2019) learns an actor-critic network using the representations learnt by an end-to-end trained world model. DreamerV2 (Hafner et al., 2021) improves the representations by replacing the Gaussian latents with reparameterized gradients (Kingma & Welling, 2013) using straight-through gradients (Bengio et al., 2013). DreamerV3 (Hafner et al., 2025) offers architectural improvements to address scaling challenges in world models for generalizing across multiple domains.

Concurrently, transformer-based world models have been developed by replacing RNN-based world models by Transformer-based architectures using self-attention to process past context, with IRIS (Micheli et al., 2022) being one of pioneers on utilizing a transformer for learning a world model. IRIS uses VQ-VAE (Kingma & Welling, 2013; Van Den Oord et al., 2017) to quantize the observations into tokens to learn a GPT2 (Radford et al., 2019) backbone for the world model. TransDreamer (Chen et al., 2022) replaces DreamerV3's RSSM with a transformer-based state-space model (TSSM) to improve world models for long-term memory tasks. Similarly, Transformer-based World Model (TWM) (Robine et al., 2023) encodes states, actions and rewards as distinct successive input tokens for the autoregressive Transformer, reconstructing input images without the world model hidden states, discarding past context temporal information. Δ-IRIS (Micheli et al., 2024) improves the efficiency by encoding the stochasticity and simulating the resulting tokens. Furthermore, STORM (Zhang et al., 2023) uses a stochastic transformer to improve the efficiency and precision of the world model. The rollout-based methods (Sutton, 1991; Kaiser et al., 2019; Yun et al., 2024; Lee et al., 2024) built over the years are generally guilty of anomalies, producing false targets that an agent learns from when unchecked, which motivates our work.

**Understanding anomalies in world models**

Talvitie (2014) is one of the earliest works that highlights the existence of anomalous states in an agent model and shows that learning value function on these can corrupt the policy learning process. Similar ideas were also discussed in Jafferjee et al. (2020). In simple benchmarks, they show hallucinated value updates with different variants of Dyna (Sutton, 1991). From a goal-reaching perspective, Di Langosco et al. (2022) empirically shows that hallucinated targets can cause an RL agent to misclassify its goals despite retaining goal-reaching capabilities. To address the problem of models producing invalid states when iterating many steps, Lo et al. (2024) proposes to use local, subgoal-conditioned policies by constraining background planning to a set of (abstract) subgoals. Recently, Zhao et al. (2025) systematically showed different ways an agent can conduct delusional planning and proposed ways using hindsight experience replay (HER) relabeling to mitigate them.

One way to better understand anomalies is by turning to insights from human perception. In particular, Hénaff et al. (2019) observe that natural visual input trajectories tend to have straighter representations—termed perceptual straightening—whereas artificially manipulated sequences exhibit pronounced curvature. This phenomena connects to world models, where anomalous rollouts can diverge from natural, plausible dynamics. This motivates us to revisit this idea of in the context of MBRL, treating hidden state from observation-action trajectories as perceptual trajectory. Inspired by the hypothesis that plausible trajectories should evolve smoothly in latent space, we attempt to *straighten* the representations by imposing a linear prior on the

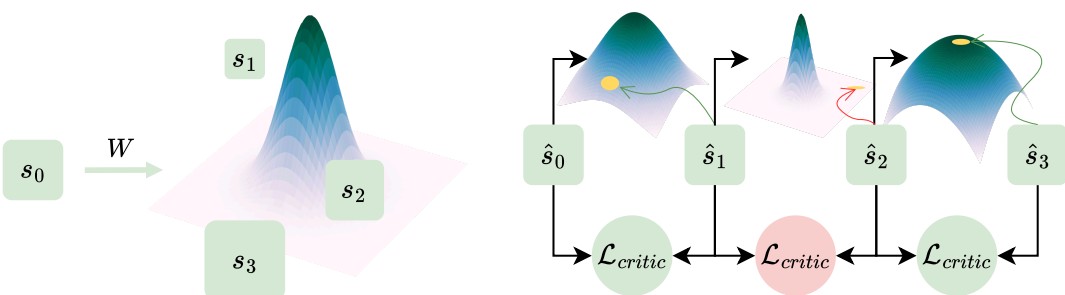

Figure 2: Illustration of LP method. *Left:* Training phase—we project $s_0$ to a Gaussian space under $W$ and minimize weighted $L_2$ distance between Gaussian samples and neighboring states ($s_{1:3}$). *Right:* Inference phase—we reject anomalous predicted transitions (e.g., $\hat{s}_1 \rightarrow \hat{s}_2$) that exceed the consistency threshold by suppressing their value updates.

them. We hope this attempt to identify anomalous states will spur further exploration of biologically inspired consistency checks in MBRL.

## 3 Method

### 3.1 Markov Decision Process (MDP)

In RL, an agent uses trial and error to explore an environment and improve its decision-making strategy. This agent interacts with the environment by taking actions and receiving rewards to learn an optimal policy that maximizes cumulative returns. This interaction is typically modeled as a *Markov Decision Process* (MDP) $\mathcal{M} = (\mathcal{S}, \mathcal{A}, P, R, \gamma)$. In the MDP, $\mathcal{S}$ denotes the state space, $\mathcal{A}$ the action space, $P(s' \mid s, a)$ the transition probability between states given an action, $R(s, a)$ the reward function, and $\gamma \in [0, 1]$ the discount factor that prioritizes immediate rewards over distant ones. The goal of RL is to learn a policy $\pi(a \mid s)$ that maximizes the expected discounted reward $J(\pi) = \mathbb{E}_\pi \left[ \sum_{t=0}^{\infty} \gamma^t R(s_t, a_t) \right]$ in order to solve problems. In pixel-based Atari, just as many real-life tasks, the agent does not receive true environment states. Rather it receives observation $o_t$ which presents an incomplete information about the underlying state. Therefore, it must form its belief over the latent states from observation to make decisions.

### 3.2 High level overview

State-space world models usually use RSSMs (Hafner et al., 2021; 2025) or TSSMs (Zhang et al., 2023; Robine et al., 2023) for differentiable dynamics, learning representations of visual inputs end-to-end through backpropagation of Bellman errors from imagined trajectories. The training procedure is usually the following. First, an encoder maps visual input $o_t$ to $z_t$. Then, a dynamics predictor predicts hidden state $h_t$ given a trajectory of $z_{<t}$ and $a_{<t}$, capturing the information before time $t$. Finally, $h_t$ predicts the next representation $z_t$, and the concatenation of these two representations predicts the reward $r_t$ and continuation $c_t$. Usually, an actor-critic agent learns the value and policy function in response to $h_t$. Thus, for the latter part of this work, we will use $s_t$ and $h_t$ interchangeably, both to mean an input state passed to the policy learner.

Having defined the basic strategy for training a world model, we now explain LP with a simple example using Figure 2. Take a state $s_0$. We define a linear transformation to a Gaussian latent space under $W$. During training, we minimize the difference between the predicted distribution and neighboring states $s_{1:3}$ by sampling states and minimizing the $L_2$ loss between them. During training the RL agent in the imagination

using world model inference, we label states as anomalous if they are *too* far from the Gaussian space linearly predicted from their previous state. Finally, we avoid updating the values of such anomalous states. In the next section, we will formalize this approach.

### 3.3 Linear consistency loss

We would like to transform the hidden state $s_t$ to be approximately linear under the linear transformation matrix $W$ such that $s_t \approx W s_{t-1}$. Thus, we define a simple loss function

$$\mathcal{L}_{consistency} = \|s_{t+1} - W s_t\| \tag{1}$$

That said, to make the linear relationship learnt more efficiently, we want to enforce linearity over more states at once, i.e., for $s_{t:t+K}$, we can define $\mathcal{L}_{consistency}$ to restrict linear projection of $s_t$ to be close to $s_{t+1:t+K}$. Thus, we define a Gaussian projection from $s_t$,

$$\mu_t, \sigma_t = W s_{t-1}$$

Finally, we revise the loss equation 1 to the following.

$$\mathcal{L}_{consistency} = \sum_{i=0}^{K-1} \mathrm{w}_i \|s_{t+i} - \hat{s_{t+i}}\|. \tag{2}$$

$$\text{where, } \hat{s_{t+1:t+K}} \sim \mathcal{N}(\mu_t, \sigma_t) \text{ and, } \mathrm{w}_i = \frac{\exp(-\delta i)}{\sum_{j=0}^{K-1} \exp(-\delta j)}$$

Here, w is the scaling parameter where $\delta$ is the decay rate that determines how quickly we *forget* farther-ahead predictions when enforcing linearity. From another perspective, this loss ensures that the linear projection of $s_{t-1}$ captures $s_{t:t+K-1}$ in the Gaussian space.

We base our implementation by incorporating the losses reported in the standard world model architecture (e.g., DreamerV3). The representation and the dynamics loss involve the KL divergence between the posterior state $z_t$ and the predicted prior state $\hat{z}_t$ with free bits (Kingma et al., 2016). Next, we train the reward predictor and decoder via the symlog loss and the continue predictor via binary classification loss. Overall, these losses are the following.

$$\begin{aligned}
\mathcal{L}_{\mathrm{pred}}(\phi) &\doteq -\ln p_\phi(o_t \mid z_t, h_t) \\
&\quad - \ln p_\phi(r_t \mid z_t, h_t) - \ln p_\phi(c_t \mid z_t, h_t) \\
\mathcal{L}_{\mathrm{dyn}}(\phi) &\doteq \max\left(1, \mathrm{KL}\left[\mathrm{sg}(q_\phi(z_t \mid h_t, x_t)) \parallel p_\phi(z_t \mid h_t)\right]\right) \\
\mathcal{L}_{\mathrm{rep}}(\phi) &\doteq \max\left(1, \mathrm{KL}\left[q_\phi(z_t \mid h_t, x_t) \parallel \mathrm{sg}(p_\phi(z_t \mid h_t))\right]\right)
\end{aligned}$$

Putting it all together, the total loss is:

$$\mathcal{L}(\phi) \doteq \mathbb{E}_{q_\phi}\left[\sum_{t=1}^{T}\left(\beta_{\mathrm{pred}}\mathcal{L}_{\mathrm{pred}}(\phi) + \beta_{\mathrm{dyn}}\mathcal{L}_{\mathrm{dyn}}(\phi) + \beta_{\mathrm{rep}}\mathcal{L}_{\mathrm{rep}}(\phi) + \beta_{\mathrm{consistency}}\mathcal{L}_{\mathrm{consistency}}(\phi)\right)\right].$$

We train the parameters of the different components of the world model together with this loss.

The above idea is related to a number of works in different contexts. In contrastive predictive coding (CPC) and its variants (Oord et al., 2018; Henaff, 2020), the representations are trained to be more predictive from one another through evolving them linearly across time-steps, effective in modalities such as image, audio, and text. Sermanet et al. (2018) applies similar idea in robotics by encouraging trained embeddings to lie along

predictable trajectories with a time-contrastive network applied on a linear-Gaussian actor. Deviating to our work but mention-worthy is Wiskott & Sejnowski (2002), which impose a *slowness* prior (linear combination of non-linear expansion) for temporally-correlated inputs, yielding a low-curvature trajectory in the latent space. A similar but differently motivated idea is Yang et al. (2021), which enforces forward and backward predictions to have low errors, essentially making the embeddings to be more invariant in the trajectory, finding its use in video representation learning.

### 3.4 Detection of anomalous environment dynamics

We label a state transition as anomalous if deviation of the state- predicted by world model from the linearity assumption is above a threshold. At inference time, we define a generated state $\hat{s}_t$ to be anomalous if:

$$\|\hat{s}_t - W\hat{s_{t-1}}\|_2^2 > \delta, \tag{3}$$

where $\delta = \mu + k_\sigma \sigma$ is a threshold derived from the mean $\mu$ and standard deviation $\sigma$ of the training-time consistency loss, and $k$ is a sensitivity constant. Throughout our work, we use $k_\sigma = 1.645$.[2]

Thresholding deviation from an agreed-upon structure has long served an effective way to detect anomalous states in different contexts. For example, in classical signal processing, a measurement is regarded as anomalous if the squared norm of a state's innovation exceeds a threshold Ding (2008). In robust principal component analysis (PCA) (Candès et al., 2011), high-dimensional states are reconstructed using near-linear combinations of a low-rank subspace and sparse signals by thresholding on a specific deviation. Number of methods detect anomaly projecting to latent space using an autoencoder (Sakurada & Yairi, 2014) or Gaussian subspace (Lee et al., 2018) and thresholding deviation on a distance metric. In a different context, diffusion models have been used to identify abnormal patterns in complex data by using a distance metric in the generative latent space as anomaly criteria (Gavrilev & Burnaev, 2023; Christopher et al., 2024).

In the following theorem 1, we make theoretical justification that using a threshold $\delta$ to detect and filter out anomalous world model states is an appropriate approach. Our anomaly detection setup depends on having a finite mean $\mu$ and variance $\sigma$. With linear prior, $\mu$ and $\sigma$ are both finite and, therefore, the false positive rate of the anomaly detection is bound. The proof can be found in §A.

Let $s_t \in \mathbb{R}^d$ be a true latent state at time $t$, and suppose a generative model learns to produce $\hat{s}_t$ given $\hat{s_{t-1}}$, i.e., $\hat{s}_t \sim p_\theta(s_t \mid s_{t-1})$. During training, we impose an additional consistency constraint enforcing approximate linearity in transitions: $\mathcal{L}_{\text{consistency}} = \mathbb{E}_t \left[\|s_t - Ws_{t-1}\|_2^2\right]$, where $W \in \mathbb{R}^{d \times d}$ is a learned linear transition matrix.

We make the following assumptions:

- *The perceptual straightening assumption*: The true hidden dynamics are approximately linear with Gaussian noise: $s_t = Ws_{t-1} + \epsilon_t, \quad \epsilon_t \sim \mathcal{N}(0, \Sigma)$.

- The generative model outputs: $\hat{s}_t = \tilde{W}\hat{s_{t-1}} + \tilde{\epsilon}_t, \quad \tilde{\epsilon}_t \sim \mathcal{N}(0, \tilde{\Sigma})$, for some matrix $\tilde{W}$ approximating $W$.

**Theorem 1** (Bounded false-positive rate of the anomaly detector). *Let $A_t := (\tilde{W} - W)\hat{s_{t-1}} \in \mathbb{R}^d$ (deterministic given $\hat{s_{t-1}}$) and*

$$X_t = \|\hat{s}_t - W\hat{s_{t-1}}\|_2^2 = \|A_t + \tilde{\epsilon}_t\|_2^2 = \sum_{i=1}^{d} \left(A_{t,i} + \tilde{\epsilon}_{t,i}\right)^2.$$

---

[2]Since $W\hat{s_{t-1}}$ defines a Gaussian projection from $\hat{s_{t-1}}$ based on Eq 2, we implement the deviation by sampling a fixed number of states from the distribution defined by $W\hat{s_{t-1}}$ and measuring mean deviation of $\hat{s}_t$ against them. This is equivalent to labeling a predicted state as anomalous if it has a low probability under the true transition kernel.

[3]Even with the *highest* average sliding-window linearity error, given this is an aggregate, all state-transitions are *not* necessarily anomalous and vice versa.

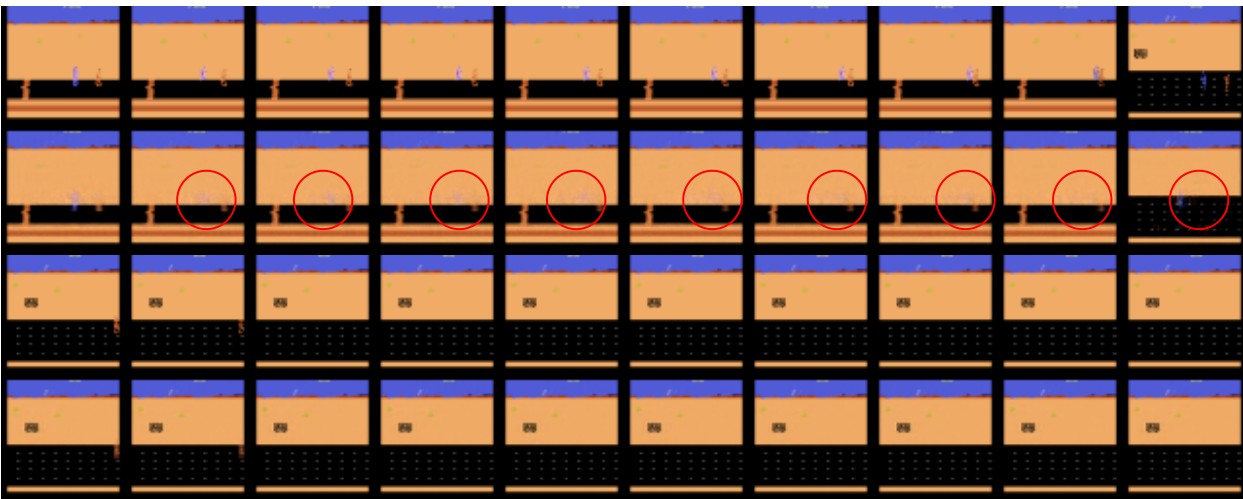

Figure 3: Decoded output trajectories from states with low vs. high linearity in the game *Road Runner*. We identify states based on their average sliding-window linearity error, selecting the most and least linear trajectories.[3] *Rows 1-2:* Ground-truth observations (row 1) and decoded outputs from low-linearity states (row 2) showing anomalies such as a vanishing chaser. *Rows 3-4:* Ground-truth observations (row 3) and decoded outputs from high-linearity states (row 4) showing consistent predictions. All trajectories are from the same training stage (approximately 25% through training) when the rejection rate is high (see Figure 9). Additional examples are shown in Figures 10 and 11.

*Under assumption 2, $X_t$ is a generalized noncentral chi-square statistic on $d$ degrees of freedom, with finite moments*

$$\mu := \mathbb{E}[X_t] = \|A_t\|_2^2 + \operatorname{Tr}(\tilde{\Sigma}), \qquad \sigma^2 := \operatorname{Var}(X_t) = 2\operatorname{Tr}(\tilde{\Sigma}^2) + 4\,A_t^\top \tilde{\Sigma} A_t.$$

*Hence, for any $k_\sigma > 0$, the false-positive rate of the rule $X_t > \mu + k_\sigma \sigma$ obeys the one-sided (Cantelli) inequality*

$$\mathbb{P}(X_t > \mu + k_\sigma \sigma) \le \frac{1}{1 + k_\sigma^2}. \tag{4}$$

The bound is distribution-free and deliberately conservative; it shows only that, *under the linear prior*, $X_t$ has finite, estimable $\mu, \sigma$, so the detector's false-positive rate is bounded and controllable by $k_\sigma$—a guarantee that fails without the prior, where $\|\hat{s}_t - W s_{t-1}\|^2$ need not concentrate. In practice we standardize $z_t = (X_t - \hat{\mu})/\hat{\sigma}$ and reject at $z_t > 1.645$; for moderate/large $d$ a standardized (noncentral) chi-square is approximately Gaussian (CLT over the $d$ coordinates), making the one-sided 95% quantile a principled operating point.

### 3.5 Removing contribution of anomalous states from policy learning

Having built the formulation to detect anomalous states reliably, we now turn our attention to removing the contribution of these states for policy learning for an actor-critic agent (Konda & Tsitsiklis, 1999) used commonly for MBRL algorithms. Since these agents rely on their value function to facilitate policy learning, we simply turn off the value updates for the anomalous state transitions. That is, for imagined rollout $s_{0:H}$ where $H$ is the imagination horizon, we turn off value updates if $\hat{s}_t \to \hat{s}_{t+1}$ is anomalous, i.e., $\mathcal{L}_{critic}(\hat{s}_t \to \hat{s}_{t+1}) = 0$.

## 4 Experiments and results

In this section, we describe our results on the popular Atari100k benchmark. First, we show that our DreamerV3-based method, LP-DreamerV3 is able to reject anomalous states. Next we apply LP on both DreamerV3 and STORM and compare them against several baselines such as GRU based DreamerV3,

transformer based TWM, IRIS, $\Delta - $ IRIS and STORM, and SimPLe. Finally, we empirically show the reasoning behind several design decisions of LP.

### 4.1 Atari100 benchmark

Atari100k benchmark (Kaiser et al., 2020) is a RL sample efficiency evaluation benchmark. It contains 26 Atari games with a budget of 100k interactions (or, 400k frames) that is equivalent to around 1.85 hours of real-time play by a human.

### 4.2 Image comparison

*Is our method able to detect anomalous states?* To answer this question, we visualize the 10-frame trajectory from 5 games with the highest and lowest linearity and compare them against the true trajectories. Figure 3, 10, and 11 shows the visualization. It is evident that trajectories that are free of any obvious anomalies are more linear (bottom rows). On the other hand, trajectories with lower linearity exhibit signs of anomalies including missing key objects (opponent in *Road Runner*, ice floe in *Frostbite*), predicting non-existent objects (fighters in *Kung Fu Master*), and replacing information (*ghost color in Ms Pacman, score in Boxing*)(top rows).

### 4.3 Full benchmark

Table 1: Agent scores and human-normalized metrics on the 26 games of the Atari 100k benchmark. We show average scores over 5 seeds. Bold numbers indicate best performing method for each game. Asterix (*) indicates baseline yielding higher score with LP.

| Game | Random | Human | SimPLe | TWM | IRIS | DreamerV3 | STORM (Rep) | Δ-IRIS | LP-DreamerV3 | LP-STORM |
|---|---|---|---|---|---|---|---|---|---|---|
| Alien | 228 | 7128 | 617 | 675 | 420 | 959 | **1364** | 391 | 1265* | 1290 |
| Amidar | 6 | 1720 | 74 | 122 | 143 | 139 | **239** | 64 | 175* | 203 |
| Assault | 222 | 742 | 527 | 683 | **1524** | 706 | 707 | 1123 | 862* | 721* |
| Asterix | 210 | 8503 | 1128 | 1116 | 854 | 932 | 865 | **2492** | 1166* | 1213* |
| Bank Heist | 14 | 753 | 34 | 467 | 53 | 649 | 375 | **1148** | 1013* | 424* |
| Battle Zone | 2360 | 37188 | 4031 | 5068 | 13074 | 12250 | 10780 | 11825 | **14153*** | 5840 |
| Boxing | 0 | 12 | 8 | 78 | 70 | 78 | 80 | 70 | **82*** | 80 |
| Breakout | 2 | 30 | 16 | 20 | 84 | 31 | 12 | **302** | 35* | 14* |
| Chopper Command | 811 | 7388 | 979 | 1697 | 1565 | 420 | **2293** | 1183 | 1772* | 1841 |
| Crazy Climber | 10780 | 35829 | 62584 | 71820 | 59324 | **97190** | 54707 | 57854 | 80881 | 55273* |
| Demon Attack | 152 | 1971 | 208 | 350 | **2034** | 303 | 229 | 533 | 322* | 186 |
| Freeway | 0 | 30 | 17 | 24 | **31** | 0 | 0 | **31** | 7* | 0 |
| Frostbite | 65 | 4335 | 237 | 1476 | 259 | 909 | 646 | 279 | **2393*** | 1495* |
| Gopher | 258 | 2412 | 597 | 1675 | 2236 | 3730 | 2631 | **6445** | 3930* | 4371* |
| Hero | 1027 | 30826 | 2657 | 7254 | 7037 | 11161 | 11044 | 7049 | 7833 | **13453*** |
| James Bond | 29 | 303 | 100 | 362 | 463 | 445 | 552* | 309 | 362 | 448 |
| Kangaroo | 52 | 3035 | 51 | 1240 | 838 | **4098** | 1716 | 2269 | 1730 | 2096* |
| Krull | 1598 | 2666 | 2205 | 6349 | 6616 | 7782 | 6869 | 5978 | **8500*** | 6478 |
| Kung Fu Master | 258 | 22736 | 14862 | 24555 | 21760 | 21420 | 20144 | 21534 | **30273*** | 25065* |
| Ms Pacman | 307 | 6952 | 1480 | 1588 | 999 | 1327 | 2673 | 1067 | **3024*** | 2534 |
| Pong | −21 | 15 | 13 | 19 | 15 | 18 | 8 | **20** | **20*** | 7 |
| Private Eye | 25 | 69571 | 35 | 87 | 100 | 882 | 2734 | 103 | 952* | **3143*** |
| Qbert | 164 | 13455 | 1289 | 3331 | 746 | 3405 | 2986 | 1444 | 1221 | **4271*** |
| Road Runner | 12 | 7845 | 5641 | 9107 | 9615 | 15565 | 12477 | 10414 | **17252*** | 8151 |
| Seaquest | 68 | 42055 | 683 | 774 | 661 | 618 | 525 | **827** | 645* | 415 |
| Up N Down | 533 | 11693 | 3350 | **15982** | 3546 | 7600 | 7985 | 4072 | 9831* | 8514* |
| Normalized Mean (%) | 0 | 100 | 33 | 96 | 105 | 112 | 95 | **139** | 121* | 96* |
| Normalized Median (%) | 0 | 100 | 13 | 51 | 29 | 49 | 36 | 53 | **55*** | 43* |

We implement LP on STORM and DreamerV3, two popular world models built on transformer and GRU, respectively and benchmark them on the Atari100k benchmark. Table 1 shows the results. Following previous works, we calculate the normalized human metrics and compare the mean and median across all games, where human normalized score is calculated with the following equation:

$$\frac{\text{agent score} - \text{random score}}{\text{human score} - \text{random score}}$$

LP-DreamerV3 improves DreamerV3's normalized human mean by 9% and median by 6%, achieving a better score than DreamerV3 in 21 games out of 26. Notably, it achieves this while rejecting 11% of value updates.

It greatly improves in games where initial anomalies plaque the value function learning such as *Frostbite*, *Ms Pacman*, and *Road Runner*.

Regrettably, the same is not true for LP-STORM, which only improves STORM's normalized human mean by 1%. This is understandable once we turn our attention to rejection rates of LP-STORM compared to LP-DreamerV3 from the Figure 9. Clearly, LP-STORM is hardly able to reject any states. There could be multiple reasons for this behavior, such as model capacity, data efficiency, or linearity in transformers (Razzhigaev et al., 2024), but future study is needed to determine the underlying cause.

> **Takeaway:** Rejecting anomalous states improves DreamerV3, but not STORM.

## 4.4 Effect of layer depth

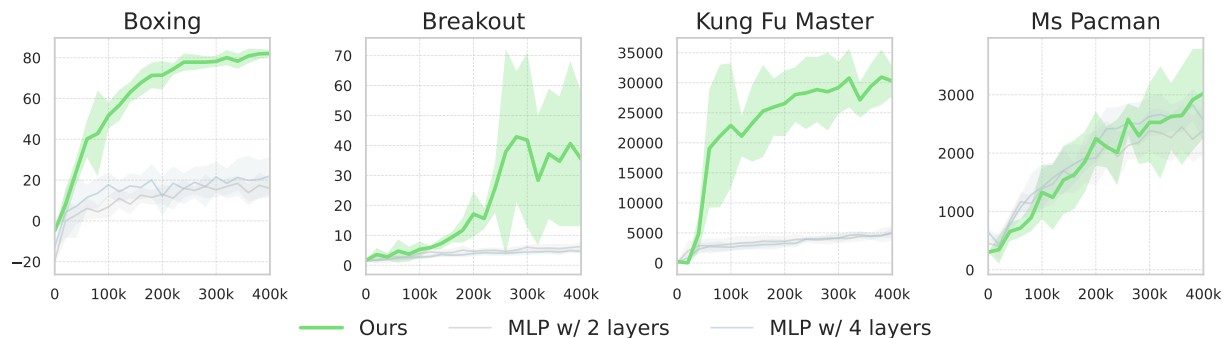

Figure 4: Is our linearity assumption *too* strong? Comparison of LP (linear projection) with MLPs with 2 and 4 layers. We see decreased performances across 4 games for more complex projections.

*Is one simple linear layer enough?* To understand whether our linear prior holds against more complex MLPs, we compare LP on 4 games with similar approach but with MLPs of 2 and 4 layers. Figure 4 shows the training curves in 5 seeds, clearly showing that LP gets stronger results. This is likely due to a deeper MLP strongly overfitting the data, rejecting most of the states.

> **Takeaway:** Deeper MLPs tend to overfit data and are ineffective for anomaly detection.

## 4.5 Effect of rejection

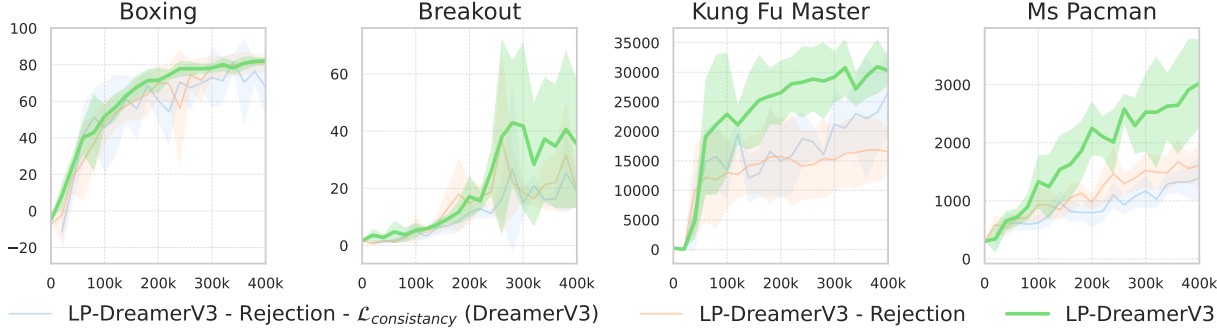

Figure 5: Comparison of LP-DreamerV3 without rejection and DreamerV3. We see performance decreases when $\mathcal{L}_{consistency}$ is added, but greatly improves when anomalous states are rejected.

*How much does value rejection play a role in training?* To test if HELP is *simply* learning better representation courtesy of the $\mathcal{L}_{consistency}$ loss, we remove value update rejection from LP-DreamerV3, i.e., DreamerV3 with $\mathcal{L}_{consistency}$, and compare it against LP-DreamerV3 and DreamerV3, i.e., LP-DreamerV3 with all modifications removed. Figure 5 shows our findings. We see that DreamerV3 with consistency loss actually worsens its performance. This is indeed understandable as a linear prior is a bottleneck. However, the performance greatly improves when we pair value update rejection with $\mathcal{L}_{consistency}$, motivating our method.

> **Takeaway:** Auxiliary objective enables false value update rejection, which is key to LP.

## 4.6 Effect of $K$

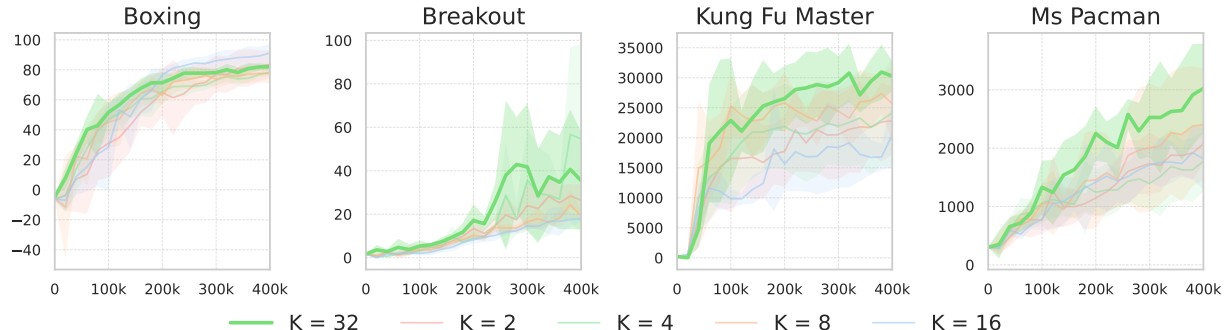

Figure 6: Comparison of LP-DreamerV3 with different choices of $K$ during training. Generally, $K = 32$ yields higher scores, motivating our choice to use $K = 32$ throughout all experiments.

*What is the optimal value of $K$?* $K$ controls the number of subsequent neighboring states we restrict the consistency loss on. We argue in eq. (1) that by encouraging more states to maintain consistency, we hope to achieve faster convergence. Figure 6 generally points to this direction, where we test LP-DreamerV3 with $K \in \{2, 4, 8, 16, 32\}$ on 4 games and see that $K = 32$ generally yields higher performances. The exception to this is *Breakout*, which greatly improves under $K = 4$.

> **Takeaway:** Promoting a larger set of states under $\mathcal{L}_{consistency}$ generally improves performance.

## 5 Discussion and limitations

In this work, we proposed a simple method—LP—to provably detect world model anomalies. Inspired by human vision, which tends to straighten neural representation of natural trajectories, LP assumes that true latent rollouts will be approximately linear and enforces a linear prior on state representations. By suppressing false value updates with LP, we show that DreamerV3 boosts human-normalized mean score by 9% while requiring less than 90% value updates with a lightweight linear prior. This prior has the potential to raise many interesting questions, e.g., *which space should be linear?, why do transformer based world models exhibit more linearity?*

Conservative by design, LP may filter out rare but valid transitions. While local linearity is often a reasonable approximation between smooth timesteps, it may prove brittle during sharp discontinuities, such as physical collisions, contacts, or abrupt environmental transitions (e.g., level changes in Atari). We believe this limitation serves as a strong motivation for *hierarchical world models* (Mattes et al., 2024; Bhirangi et al., 2024). In such architectures, high-level abstractions can "straighten" complex low-level dynamics, effectively handling discontinuities that are beyond the scope of a single-level linear prior. One other potential solution, especially in a low-data regime, is a dynamics-adaptive threshold—a promising direction we will tackle in a future work. Beyond this, we believe LP can offer important insight for training large foundation world models (e.g., Genie 1, 2 (Bruce et al., 2024; Parker-Holder et al., 2024)). Limited computational resources prevented our extended experiments, and we invite the community with greater capacity to investigate further.

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

# A   Proof of Theorem 1

**Theorem 1** (Bounded false-positive rate). Let $A_t := (\tilde{W} - W)\hat{s_{t-1}}$ and $X_t = \|\hat{s}_t - W\hat{s_{t-1}}\|_2^2 = \|A_t + \tilde{\epsilon}_t\|_2^2$. Under assumption (A2), $X_t$ has finite mean $\mu$ and variance $\sigma^2$ (below), and for any $k_\sigma > 0$ the false-positive rate of the rule $X_t > \mu + k_\sigma\sigma$ satisfies $\mathbb{P}(X_t > \mu + k_\sigma\sigma) \leq \dfrac{1}{1 + k_\sigma^2}$.

*Proof.* **Residual.** From the generative dynamics assumption,

$$r_t := \hat{s}_t - W\hat{s_{t-1}} = (\tilde{W} - W)\hat{s_{t-1}} + \tilde{\epsilon}_t = A_t + \tilde{\epsilon}_t, \qquad \tilde{\epsilon}_t \sim \mathcal{N}(0, \tilde{\Sigma}),$$

so $r_t \sim \mathcal{N}(A_t, \tilde{\Sigma})$ and $X_t = \|r_t\|_2^2 = \sum_{i=1}^d (A_{t,i} + \tilde{\epsilon}_{t,i})^2$ is a quadratic form in a $d$-dimensional Gaussian (a *generalized chi-square*). Under isotropic noise $\tilde{\Sigma} = \tilde{\sigma}^2 I$ it reduces to a scaled noncentral $\chi_d^2$ with noncentrality $\|A_t\|_2^2 / \tilde{\sigma}^2$. For the bound we use only its first two moments.

**Moments.** Expanding $X_t = A_t^\top A_t + 2A_t^\top \tilde{\epsilon}_t + \tilde{\epsilon}_t^\top \tilde{\epsilon}_t$ and using $\mathbb{E}[\tilde{\epsilon}_t] = 0$, $\mathbb{E}[\tilde{\epsilon}_t^\top \tilde{\epsilon}_t] = \mathrm{Tr}(\tilde{\Sigma})$,

$$\mu := \mathbb{E}[X_t] = \|A_t\|_2^2 + \mathrm{Tr}(\tilde{\Sigma}).$$

Since $\mathrm{Var}(2A_t^\top \tilde{\epsilon}_t) = 4A_t^\top \tilde{\Sigma} A_t$, $\mathrm{Var}(\tilde{\epsilon}_t^\top \tilde{\epsilon}_t) = 2\,\mathrm{Tr}(\tilde{\Sigma}^2)$, and $\mathrm{Cov}(2A_t^\top \tilde{\epsilon}_t, \tilde{\epsilon}_t^\top \tilde{\epsilon}_t) = 0$ (an odd central moment of a zero-mean Gaussian),

$$\sigma^2 := \mathrm{Var}(X_t) = 2\,\mathrm{Tr}(\tilde{\Sigma}^2) + 4A_t^\top \tilde{\Sigma} A_t.$$

**Tail bound.** With finite mean $\mu$ and variance $\sigma^2$, the one-sided Chebyshev (Cantelli) inequality gives, for any $k_\sigma > 0$,

$$\mathbb{P}(X_t > \mu + k_\sigma\sigma) = \mathbb{P}(X_t - \mu \geq k_\sigma\sigma) \leq \frac{1}{1 + k_\sigma^2},$$

which bounds the detector's false-positive rate. Without the linear prior $A_t$ and $\tilde{\Sigma}$ need not stay small/controlled, so $\mu, \sigma$ are not tightly estimable and the guarantee is lost. □

# B    Training curves for Dreamer-V3

Here we provide the training curves of DreamerV3 compared to LP-DreamerV3 for the Atari100k benchmark.

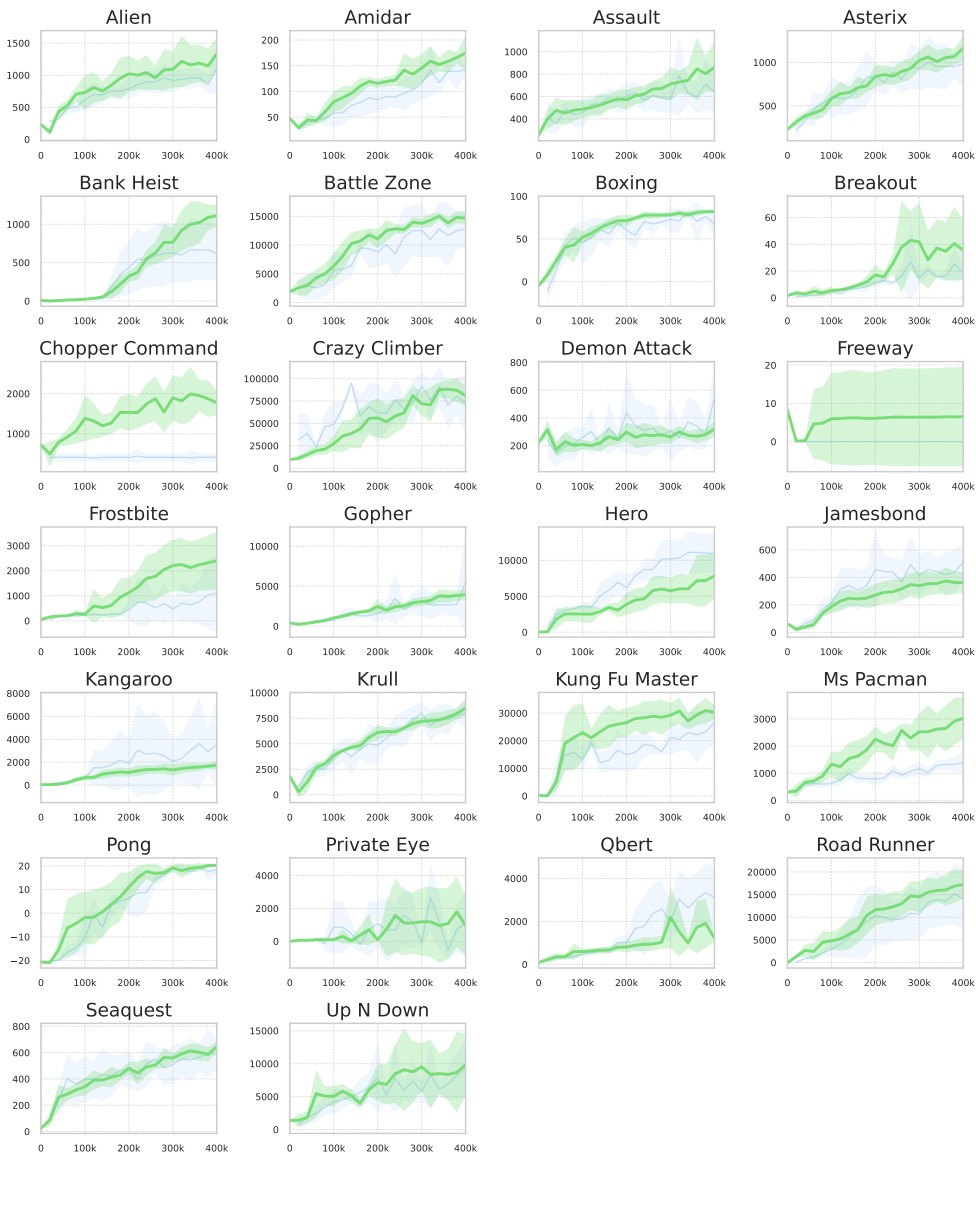

Figure 7: Training profiles across all the checkpoints for the Atari 100k benchmark for LP-DreamerV3 and DreamerV3. The solid line represents the average over 5 seeds while the fill area is defined in terms of maximum and minimum values corresponding to each checkpoint.

## C    Training curves for STORM

Here we provide the training curves of STORM compared to LP-STORM for the Atari100k benchmark.

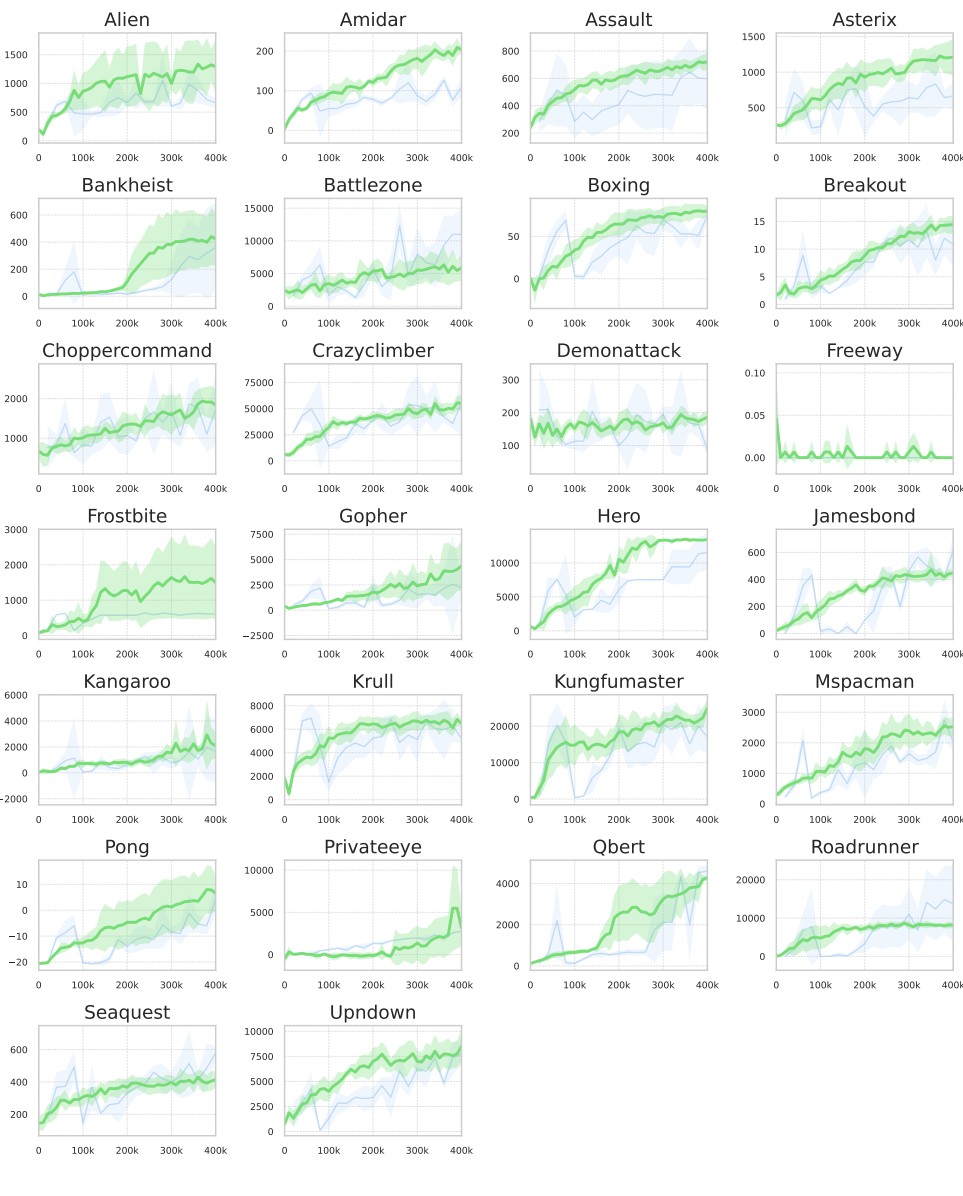

Figure 8: Training profiles across all the checkpoints for the Atari 100k benchmark for LP-STORM and STORM. The solid line represents the average over 5 seeds while the fill area is defined in terms of maximum and minimum values corresponding to each checkpoint.

## D    Rejection curves

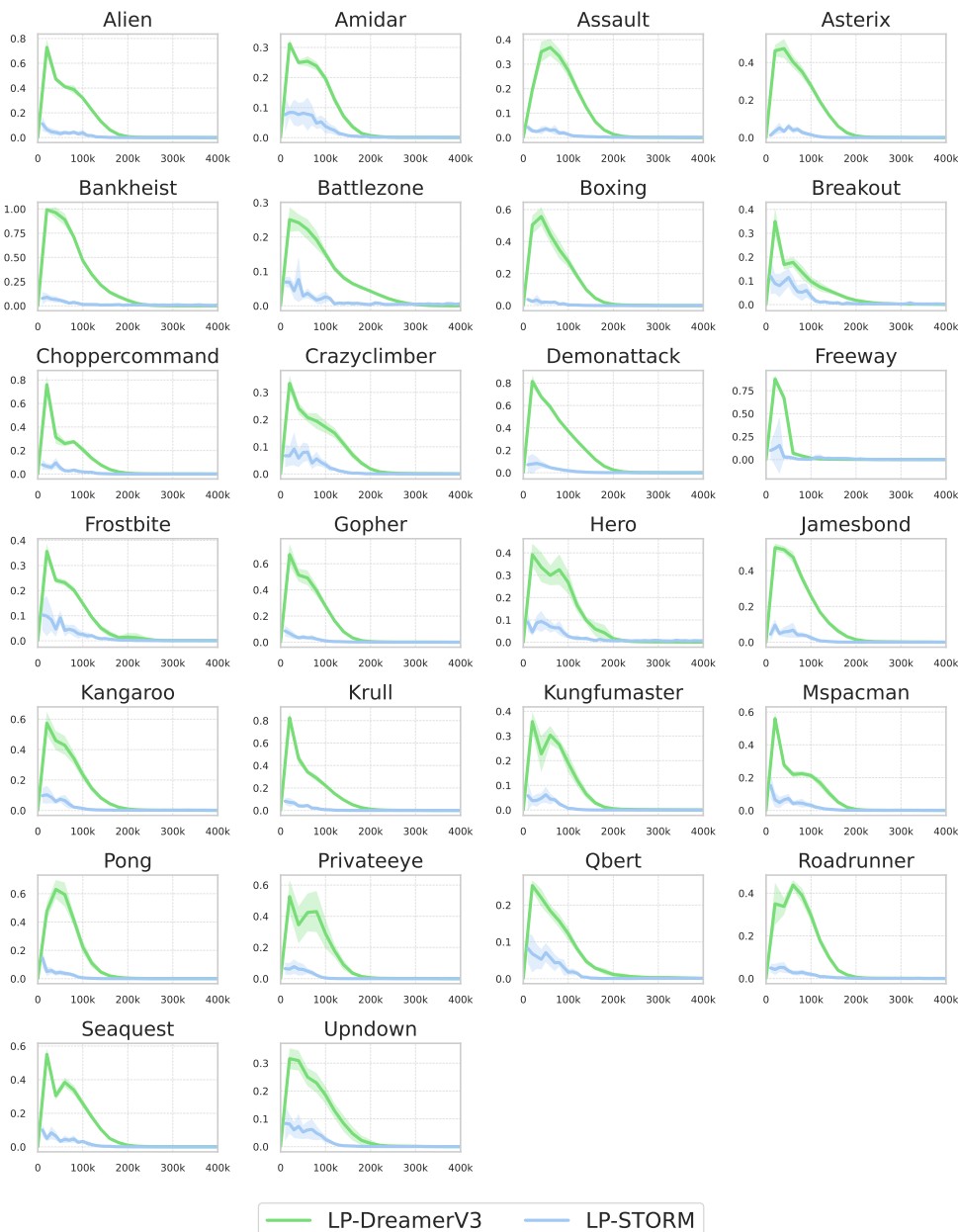

Figure 9: Rejection rates across all training stages for the Atari 100k benchmark for LP-DreamerV3 and LP-STORM. The solid line represents the average over 5 seeds while the fill area is defined in terms of maximum and minimum values corresponding to each training stages. It shows DreamerV3 has higher rejection rates compared to STORM.

# E Additional visuals

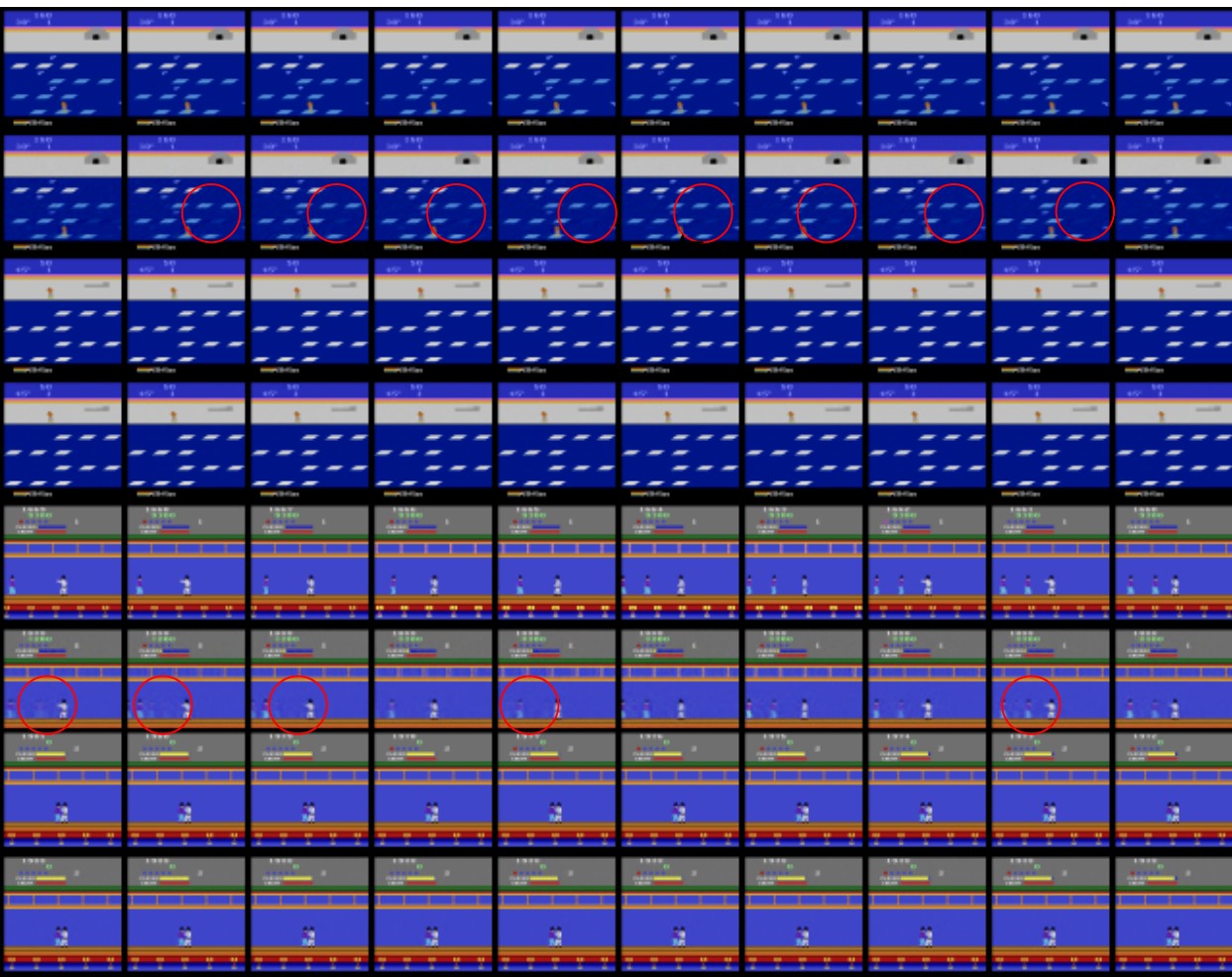

Figure 10: More visuals from the setup described in the Figure 3. Across the games, we show that states with poor linearity show signs of delusion, such as extra ice floe in *Frostbite*, and ghost fighter in *Kung Fu Master*. In comparison, states with high linearity generally show no signs of such irregularities.

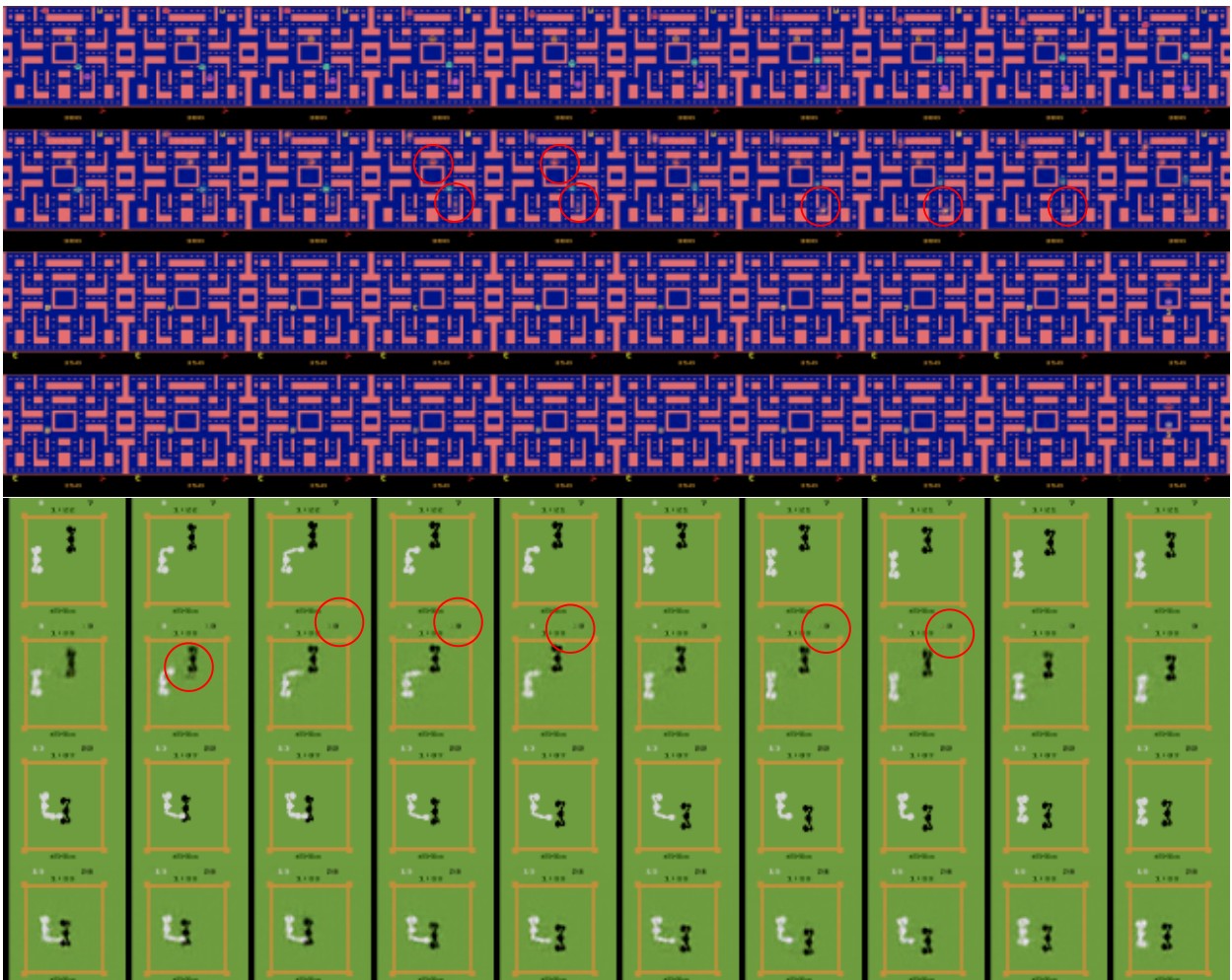

Figure 11: More visuals from the setup described in the Figure 3. Across the games, we show that states with poor linearity show signs of delusion, such as vanishing/color-changing ghosts in *Ms Pacman*, and blurry opponent or inconsistent scoring in *Boxing*. In comparison, states with high linearity generally show no signs of such irregularities.

# F    Hyperparameters

For LP-DreamerV3 and LP-STORM, we follow the default hyperparameters used in the original implementation in Hafner et al. (2025) and Zhang et al. (2023). Only hyperparameters original to LP is $K$ and $\delta$. We present them in the table 2.

Table 2: Hyperparameters

| | | |
|---|---|---|
| Decay rate | $\delta$ | 0.5 |
| Number of subsequent neighbors | $K$ | 32 |

## G  Hardware

We relied on NVidia A100 GPUs for all our experiments. There is no additional time budget required for our method built on DreamerV3 and STORM compared to the base method. Overall, we used 5 seeds for all experiments, requiring  150 A100 days.

## H  A note on state space and observation space

The likes of DreamerV3 and STORM models predict world observation $\hat{o_t}$ sampling from a learnt probability distribution, i.e., $\hat{o_t} \sim p_\phi (o_t | z_t, h_t)$, where $z_t$ is the predicted observation embedding and $h_t$ is the hidden state of dynamics backbone. This technicality renders qualitative evaluation of predicted rollouts in the pixel space weaker for two reasons. First, since there can be many $h_t$ pointing to the same $\hat{o_t}$, a perfect reconstruction does not necessarily redeem $h_t$ free from anomalies. In the same way, as one $h_t$ can produce multiple different $\hat{o_t}$ because of probabilistic reconstruction, a seemingly deviated pixel reconstruction *may* still come from a true state. From a latent representation learning perspective, assuming a true latent $z*$, if for learnt latent $z \approx z*$ and $z' \approx z*$, $||z' - z*|| < ||z - z*||$, it could be that $L_{recon}(z') > L_{recon}(z)$. This being said, in practice, stochasticity of prediction generally shows little impact for Atari100k. Thus, and in absence of a better alternative, we show anomaly detection examples with their respective decoder outputs.

