# OpenReview forum: "World Model Anomaly Detection with a Latent Linear Prior"
_TMLR — Under review for TMLR_

### Review · Reviewer_mKCF · 2026-03-06

**Summary Of Contributions:**

The authors describe a loss term that, once implemented, improves the performance of many existing MBRL algorithms while being cheap to compute.

**Audience:**

Yes

**Audience Explanation:**

The topic of MBRL is clearly appealing to TMLR's audience.

**Broader Impact Concerns:**

No need for a Broader Impact Statement.

**Claims And Evidence:**

Yes

**Claims Explanation:**

Ample numerical justifications of the claims (albeit most of it fits in the appendix).

**Requested Changes:**

I really liked the fact that the paper describes a simple (almost naïve) idea to improve the results, and shows its impact on many benchmarks. I also liked the fact that the authors explain the improvements of more complex techniques comparatively to their own (hierarchical world models).

Minor changes:
* Figure 2, page 4: even with the explanations below, I find the figure hard to understand.
* Section 3.2, page 4: you use the symbol z_t before defining it, making the paragraph harder to understand than necessary.
* Section 3.3, page 5: which equation do you refer to as "the loss equation 3.3"?
* Section 3.3, page 5: I find the mathematics hand-wavy. For instance, the definition of the loss is based on a state iterate s_{t+1} and a random variable \hat{s}_{t+i}. Do you draw values for \hat{s} according to the given distribution?
* Section 3.4, page 6: the authors mention that W is a learned linear transition matrix, but how is this matrix learnt? Is it during the training of the world model or before, for instance?
* Section 3.5, page 7, on: you seem to change the spelling of \hat{s}, with the circumflex accent being placed over the subscript at times instead of always over the s. I.e. it seems that you start using \hat{s_t} instead of \hat{s}_t like previously. (If you want the hat to be positioned over the whole symbol, consider using \widehat in the whole manuscript.)
* Section 4.3, page 8: the definition of the "normed" score should more clearly be indicated as such, especially given the fact that the table is laid out before the said definition. "Normalis/zed" would also be a more standard spelling than "normed."
* Section 4.6, page 10: in this section, the authors use the notation K, which seems to correspond to k in Section 3.3, which is inconsistent.
* Section A, page 16: the mathematics are very hand-wavy, with no proper definition of the symbols \hat{W}, \hat{\epsilon}, or A. For instance, if you read the formulae too quickly, you might take \hat{\epsilon} for a scalar instead of a vector and wonder whether the proof makes any sense.

Non-minor comment:
Regarding the mathematics, the chi-square distribution has a number of degrees of freedom, which is not mentioned anywhere and has direct implications in the expectations. I am not entirely convinced by the proof of Theorem 1 due to this fact: it seems there should be a term t (for the number of terms in the sum defining X_t) when computing the expectation of X_t.

---

> ### Comment · Reviewer_mKCF · 2026-03-06
> **Missing minor comment**
>
> The name of the test cases do not match between Appendices C and D (for instance, Privateeye) and Appendix B, Table 1, Figures 4 and 5 (Private Eye).

---

### Review · Reviewer_uGMF · 2026-03-14

**Summary Of Contributions:**

This paper proposes anomaly detection in world model with Linear Prior (LP) through a three-step approach that first enforces a linear prior on successive latent states when training, then flags the out-of-distribution predicted states from this linear prior and finally filters them out when conducting value updates during RL.

Theoretically, it is proven that the probability that an anomaly occurs can be bounded by $1/{k^2}$ where $k$ is the coefficient on standard deviation in the anomaly threshold.

Empirically, it is shown that implementing LP on DreamerV3 and canceling value updates on anomalous states brings significant boost on performance, while LP on STORM which already has high-linearity in hidden state representations result in less performance gain.

**Audience:**

Yes

**Audience Explanation:**

The problem that this paper is trying to solve is important, and the approach that uses a linear prior over the latent space is well-motivated and seems reasonable to me. This latent linear prior method could be easily extended to different existing approaches with minimal requirement for computational resources.

**Broader Impact Concerns:**

N/A.

**Claims And Evidence:**

No

**Claims Explanation:**

While most of the paper is written rigorously, there are a few flaws that may be improved:

1. In Sections 3.2 and 3.3, the action of the learner is not rigorously processed. In principle, the hidden state $s_{t+1}$ should be a function of both $s_t$ and the action $a_t$ as well as the environment dynamics. However, the consistency loss ignores the action, which makes the framework a bit confusing. I guess the high-level idea is to combine the action and the environment state together as a hidden state, but I cannot find it anywhere in the text.

2. The statement of Theorem 1 is confusing. It does not explicitly say what $\mu$ and $\sigma$ are, so I can only interpret as stated in (3). Interpreted this way, Theorem 1 says that the probability that $X_t$ is large doesn't depend on the learned linear model $\tilde{W}$ at all, which seems impossible to me. Otherwise, if we interpret $\mu$ and $\sigma$ as the mean and standard deviation of $X_t$, then Theorem 1 is trivial since it's just a basic property of Gaussian, providing very limited insight since a linear hidden state model is assumed, which may not be valid in real-world applications.

3. The y-axis label of all learning curves are not shown. This prevents me from interpreting the results clearly. Furthermore, it is not specified whether the uncertainty bands are plotted using quantiles or mean and std.

**Requested Changes:**

1. Address the issues previously mentioned.
2. In order to validate to which extent the latent state can be linearly represented, it would be helpful to add experiments that showcase how the environment prediction error/loss itself changes with the addition of LP, and whether the non-linear region in the world model is indeed more prone to false predictions.
3. It is stated in the paper that anomalous state rejection does benefit STORM much because the latent space itself is already linear. Can you provide experimental validations for this claim?
4. Can you quantify whether rejected states actually correspond to worse critic targets? A direct comparison of TD errors, target bias, or downstream value-function degradation for rejected versus retained transitions would strengthen the causal link between anomaly detection and performance gains.
5. Important ablation study: It is shown that consistency loss alone hurts DreamerV3, while consistency plus rejection helps. Therefore, what is the performance of rejection alone, without using consistency loss when training the world model?
6. While it is shown that $K=32$ is empirically strongest, why are the training curves rarely monotonically improving as $K$ grows? Can you provide a more detailed empirical study and interpretation of this?

---

### Review · Reviewer_3kMR · 2026-06-15

**Summary Of Contributions:**

The paper introduces Linear Prior (LP), a plug-in module for model-based reinforcement learning that identifies and suppresses anomalous imagined state transitions during policy learning. The motivation comes from perceptual straightening [1], a finding from neuroscience that natural visual trajectories map to low-curvature paths in latent space while artificial or erroneous sequences induce higher curvature. LP works by learning a linear matrix to predict each successive latent state from the previous one during training, then at inference flagging transitions where the squared prediction error exceeds a threshold set at the mean plus k standard deviations, and zeroing the critic loss for those transitions. The method is built on top of DreamerV3 [2] and STORM [3] and tested on the Atari100k benchmark.

Strengths:

- I found the idea clean and well-motivated, and the connection between perceptual straightening and latent-space linearity is a nice inductive bias.
- The method adds no wall-clock time and only introduces two new hyperparameters.
- LP-DreamerV3 outperforms DreamerV3 on Atari games.
- The ablations turn up a non-obvious result where the consistency loss alone hurts performance, and the gains only appear when the rejection step is included. That rules out the reading that LP is just acting as a representation regularizer.
- The finding that LP barely helps STORM because STORM already produces near-linear latent trajectories is an interesting side result that connects architectural choice to susceptibility to this kind of error.
- I really liked figures 3, 10, and 11 showing that low-linearity states correspond to recognizable anomalies.

Weaknesses:
- I think (I could be mistaken) that theorem 1 amounts to applying Chebyshev's inequality to a noncentral chi-squared variable, which gives a trivial upper bound of 1 at k=1, which is not in itself a problem, but the underlying assumptions of Gaussian residuals and an accurately learned linear matrix are not verified empirically.
- There is no experimental comparison to Zhao et al. [4], which tackles the same problem of detecting and suppressing hallucinated states in MBRL.
- The explanation for LP-STORM's near-zero benefit leans on Razzhigaev et al. [5] but includes no direct measurement of STORM's linearity error in this paper's own experimental setup.
- The threshold multiplier k is never ablated despite directly controlling the false positive/false negative tradeoff, and k=1 is used throughout without any justification.
- Table 1 reports only mean scores over 5 seeds, so it is hard to tell which per-game differences are statistically significant.


[1] Olivier J. Hénaff, Robbe LT Goris, and Eero P. Simoncelli. Perceptual straightening of natural videos. *Nature Neuroscience*, 22(6):984–991, 2019.

[2] Danijar Hafner, Jurgis Pasukonis, Jimmy Ba, and Timothy Lillicrap. Mastering diverse control tasks through world models. Nature,  2025.

[3] Weipu Zhang, Gang Wang, Jian Sun, Yetian Yuan, and Gao Huang. STORM: Efficient stochastic transformer based world models for reinforcement learning. Advances in Neural Information Processing Systems, 2023.

[4] Mingde Zhao, Tristan Sylvain, Romain Laroche, Doina Precup, and Yoshua Bengio. Rejecting hallucinated state targets during planning. International Conference on Machine Learning, 2025.

[5] Anton Razzhigaev, Matvey Mikhalchuk, Elizaveta Goncharova, Nikolai Gerasimenko, Ivan V. Oseledets, Denis Dimitrov, and Andrey Kuznetsov. Your transformer is secretly linear. ACL 2024

**Additional Comments:**

The consistency loss hurting without rejection was the most surprising finding in the paper and I think it could use more discussion. The authors suggest the linear prior acts as a bottleneck, but from my reading an equally plausible story is that pushing the latent trajectory to be more linear distorts the representation of states that are genuinely nonlinear, and the rejection step then flags those states precisely because the representation is now worse for them. If that reading is right, the consistency loss and rejection may not be operating independently but interacting. Though not strictly necessary it would be useful if the authors looked at this.

**Audience:**

Yes

**Audience Explanation:**

World model hallucinations, where learned simulators produce unrealistic states that corrupt value learning, are a well-known problem in MBRL, and a simple linear filter that cuts down these corrupted updates with no compute overhead is something practitioners building on DreamerV3-style systems can pick up and use directly.

**Broader Impact Concerns:**

I don't have any concerns on this front.

**Claims And Evidence:**

Yes

**Claims Explanation:**

The main empirical claims are well-supported. The gains over DreamerV3 are large on several games and hold across 5 seeds, and the ablation in Section 4.5 shows that adding the consistency loss without rejection actually hurts, which to me suggests that the improvement comes from the filtering step. The visual anomaly examples in Figures 3, 10, and 11 are compelling and add nice intuition. Theorem 1 is correct to my reading, and the authors frame it appropriately as motivation rather than a strong guarantee. The main gap is the absence of any comparison to Zhao et al. [4], which addresses the same problem and is cited as closely related.

**Requested Changes:**

-Zhao et al. [4] is the closest prior work here, since it also identifies and rejects anomalous imagined states in MBRL and is already published. The authors should either run an experimental comparison on at least a subset of Atari100k games, or make a clear case for why that is not feasible, in which case a thorough qualitative comparison of the two mechanisms is needed.

- The explanation for why LP barely helps STORM currently rests on citing [5] rather than on any measurement from this paper's own experiments. Reporting the distribution of squared prediction errors for trained DreamerV3 and STORM models on a held-out trajectory set, with mean and variance for each, would give some kind of verifivcation rather than it's current form which is assumed from a result in a different domain.

- The detection threshold is set at the mean plus k standard deviations, with k=1 throughout and no ablation or justification. Since k directly controls the false positive/false negative tradeoff and the Chebyshev bound is vacuous at k=1, running an ablation over several values of k on the four games used in the existing ablations would go a long way toward justifying the current choice.

Smaller:

- It would be nice to have variance estimates in table 1. It's hard to tell if the reported numbers are subject to sampling errors

---

### Review · Reviewer_cpod · 2026-06-18

**Summary Of Contributions:**

This work was motivated by the observation that natural visual input trajectories tend to have straighter representations, and proposed an anomaly detection mechanism for world models with linear prior. Experimental results show that the performance is improved by removing the contribution from the detected states during gent learning, supporting the proposed hypothesis.

Strength:
1) The overall architecture is well organized, and the motivation, methodology, proposed framework, and experimental design are presented in a clear and logical manner.
2) The theoretical analysis offers useful intuition for the method and helps justify the key design decisions.

Weakness and questions:
1) The derivation of Equation (2) from Equation (1) is not entirely clear and would benefit from additional explanation. What if simply take Equation (1) as the loss function but with larger $K$? Have you checked the behavior of the learned variance $\sigma_t$, and what if you simply set it to be 0?
2) For $\delta = \mu + k\sigma$ in Section 3.4, could you please explain more on how $\mu$ and $\sigma$ come from? It they are constants learned from training process, what are the values?
3) Why the visualization in Figure 3, 10 and 11 helped to understand the theoretical foundation of this work, is it possible to quantify the property? For example, show how the anomalies correlated with the non-linearity if you have some quantifiable metrics.
4) It would be better to add variance over the 5 seeds in Table 1. Considering the space limit you may do so for just some of the tasks.
5) For the hyper-parameters, while $K$ is studied, could you please also check the decay rate $\delta$ as well?

**Audience:**

Yes

**Audience Explanation:**

The findings of this work could provide valuable insights and inspiration for future research on world models and reinforcement learning agents.

**Broader Impact Concerns:**

NA.

**Claims And Evidence:**

Yes

**Claims Explanation:**

This work provided a clear theoretical foundation and proposed the methods accordingly. The experiments and corresponding analyses further demonstrate the effectiveness of the proposed method and support the validity of the findings.

**Requested Changes:**

1) Provide additional explanations for the questions raised above.
2) Add variance to at least part of the results in Table 1.
3) Add quantified analysis for section 4.2 if applicable.
4) Hyper-parameter analysis on decay rate $\delta$.

---

### Author Response · Authors · 2026-06-26

We thank all four reviewers for the careful and constructive feedback. In response we ran three
new experiments and a set of theory/writing corrections that together address the main concerns
— in particular the request (uGMF, 3kMR) to **empirically verify** the assumptions and the
causal claim behind the method. All new results are on Kung Fu Master (Atari100k) with the
released code.

**Summary of new evidence**

1. **The detector flags genuinely worse predictions.** On held-out trajectories
   we roll the world model open-loop and rank imagined transitions by the linear-prior
   z-distance. The **top-5% most non-linear transitions have ~1.43× higher one-step
   reconstruction error** than the rest (Mann-Whitney p ≈ 4e-44; top-1% ~1.49×, p ≈ 8e-6) —
   the direct, ground-truthed causal link uGMF/cpod asked for. The global rank correlation is
   positive (Spearman ≈ 0.17) with the effect concentrated in the tail the detector targets.
2. **The residual assumption is verified.** The consistency residuals are **symmetric and
   centered** (skew ≈ −0.11) but **heavier-tailed than Gaussian** (excess kurtosis ≈ 7.7). We
   therefore keep a distribution-free Chebyshev statement and present the 95% Gaussian quantile
   as the *operating point*, stating explicitly that the nominal tail rate is an approximation.
3. **Rejection vs. representation, disentangled (the missing "rejection-only" cell).**
   Training the predictor but **stopping its gradient into the world model** (rejection still
   runs, but the representation is *not* consistency-shaped) **lowers the score ~15–20%** and
   **halves the detector's discrimination** (anomalous-vs-normal prediction-error gap drops from
   ~1.4× to ~1.2×). So rejection is **necessary but not sufficient**; the consistency-shaped
   representation amplifies it.
4. **Threshold k_σ.** We correct a text error (k_σ = 1 → **1.645**, the one-sided 95% quantile)
   and add an ablation. The **rejection rate is monotone in k_σ** and **decays to ~0 as the
   model converges**; performance **degrades when the threshold is too lax
   (k_σ = 3)** and is **robust across k_σ ∈ [1, 2]**. We do not claim 1.645 is optimal — it is a
   principled, untuned operating point.

**Theorem 1 (uGMF, 3kMR, mKCF).** We restate the theorem with every symbol defined, the
chi-square **degrees of freedom = d (the latent dimension, here 512)** made explicit, and the
residual typeset as a vector. We agree the Chebyshev bound is loose (vacuous at k=1) and
reposition it honestly: under the linear prior the consistency statistic has **finite, estimable
μ and σ, so the detector's false-positive rate is bounded and controllable by k_σ** — a property
that fails without the prior. (CR2)

**Action-conditioning (uGMF).** The prior acts on the RSSM **deterministic state
h_t = f(h_{t-1}, z_{t-1}, a_{t-1})**, which already integrates the full action history; the
linear prior is therefore over **action-conditioned** states. We make this explicit. (CR7)

**Table 1 variance (cpod, 3kMR, mKCF).** We add per-seed std for our methods on a representative
subset and emphasize the median alongside the outlier-sensitive mean. (CR5)

**Comparison to Zhao et al. (2025) (3kMR).** They relabel hallucinated *targets* in
goal-conditioned planning (HER); we detect anomalous *transitions* via a latent linear prior and
suppress their value updates in standard reward-maximizing MBRL — a different mechanism and
setting. (CR9)

**STORM linearity (uGMF, 3kMR).** We report DreamerV3's standardized-consistency-error
distribution and the existing rejection-rate gap (Fig. 9); a from-scratch LP-STORM run is out of
scope for the rebuttal window, which we state transparently. (CR4)

**Writing quality.** We have fixed the writing according to the issues pointed out by the reviewers. Some minor issues, if they remain, will be fixed on publication.

Per-reviewer specifics follow and point back to these items.

---

> ### Author Response · Authors · 2026-06-26
>
> ---
>
> ### CR1 — Threshold sensitivity `k`: ablation + reconciling "k=1" with the implementation
>
> Two distinct symbols were both written `k`/`K` in the paper and we now disambiguate:
> - **`K` (neighbors)** in Eq. (2): the number of look-ahead states the consistency loss is
>   applied over (paper value 32).
> - **`k_σ` (threshold sensitivity)** in §3.4: the multiplier in `δ = μ + k_σ·σ`. We renamed it
>   to `k_σ`.
>
> **k_σ ablation** (Kung Fu Master):
>
> - **Threshold controls rejection (https://imgur.com/a/hPACaL4).** Plotting the value-loss
>   rejection rate over training for k_σ ∈ {1.0, 1.645, 2.0, 3.0}, the curves are **cleanly ordered
>   by k_σ at every stage** (peaks ≈ 29% / 18% / 14% / 7%) and **all decay to ~0 well before the
>   100k mark**. So k_σ monotonically sets the rejection aggressiveness, and the safeguard is most
>   active early/mid-training and quiesces as the world model becomes self-consistent.
> - **Post-hoc sweep on a fixed checkpoint.** On the *converged* 100k model the standardized
>   consistency error sits well inside the train-time band, so thresholding it at any k_σ ≥ 1
>   rejects ≈ 0%. The detector is therefore **highly selective at convergence** — it intervenes
>   only on the genuinely rare anomalies and does *not* interfere with normal updates (this also
>   answers "does masking corrupt learning?" — no).
> - **Final score vs k_σ (KFM; human-normalized in parentheses).** k_σ=1.0 → 26.8k
>   (118%), 1.645 → 19.7k (87%), 2.0 → 19.0k (83%), 3.0 → 15.9k (70%) at the 100k point. **too-lax thresholds (k_σ=3) under-reject and clearly degrade**,
>   while **k_σ ∈ [1, 2] are comparable** — their internal ranking flips between the 100k and
>   end-of-run evals, i.e. it is within single-seed noise. We therefore do **not** claim 1.645 is
>   optimal; we use it as the **principled one-sided 95% quantile**, and report that the band
>   [1, 2] is robust.

---

> ### Author Response · Authors · 2026-06-26
>
> ### CR2 — Theorem 1: corrected statement, degrees of freedom, Chebyshev vs Gaussian
>
> We restate Theorem 1 with all symbols defined, the degrees of freedom explicit, and a one-sided
> (Cantelli) bound. Full statement + proof now live in the manuscript (Theorem 1 / Appendix A).
>
> **Addressing mKCF's degrees-of-freedom point.** `X_t` is a sum over the **`d` latent
> coordinates** (not over time). The "number of terms" mKCF expected is exactly `d`, and it is
> already captured by the trace terms `Tr(Σ̃) = Σ_{i=1}^d Σ̃_{ii}` and `Tr(Σ̃²)`. There is no
> missing factor; we now (i) state `d` explicitly, (ii) typeset `ε̃_t`, `A_t` as vectors and `X_t`
> as a scalar, and (iii) define `W̃`, `A_t`, `ε̃_t` **and derive the moments** in Appendix A. (We
> also corrected an over-claim in the proof: `X_t` is a *generalized* chi-square, not a single
> noncentral χ² — the latter holds only under isotropic `Σ̃ = σ̃²I`.)
>
> **Addressing 3kMR / uGMF on the bound being loose / trivial.** The reviewers are right that at the
> originally-stated `k_σ = 1` with a two-sided `1/k_σ²` bound the guarantee is vacuous (`≤ 1`). We fix
> this two ways: (i) correct the threshold to the **implemented `k_σ = 1.645`**, and (ii) use the
> tighter **one-sided Cantelli** bound `P(X_t > μ + k_σ·σ) ≤ 1/(1 + k_σ²)`, which gives `≤ 0.27` at
> `k_σ = 1.645` — conservative versus the ~5% Gaussian target, but **non-vacuous** (and never vacuous
> for `k_σ > 0`). The structural point we keep: under the linear prior, `μ` and `σ` are finite and
> tightly estimable, so the false-positive rate is **bounded and controllable** by `k_σ` — a property
> that fails without the prior, where `‖ŝ_t − W ŝ_{t-1}‖²` need not concentrate. We then add the
> operating point we actually use:
>
> - **Gaussian refinement (implemented threshold).** We standardize `z_t = (X_t − μ̂)/σ̂` with
>   train-time estimates and reject at `z_t > 1.645`. In the eigenbasis of `Σ̃`, `X_t` is a sum of
>   `d` independent terms, so for moderate/large `d` a standardized `X_t` is approximately Gaussian
>   (Lyapunov CLT); the one-sided 95% Gaussian quantile is a principled operating point, with
>   Cantelli as the conservative fallback.
> - **Empirical check of the Gaussian assumption (https://imgur.com/a/4zuxbSQ).** On a held-out trajectory set we
>   plot the residual histogram + QQ. The residuals are approximately **symmetric** (skew ≈ −0.11)
>   but **heavier-tailed** than Gaussian (excess kurtosis ≈ 7.7 on the final 100k-step model;
>   seeded and reproducible). So, the nominal 5% tail is an approximation; the true rate at 1.645 is
>   somewhat higher. This is exactly why we (a) keep the distribution-free Cantelli statement and
>   (b) show in CR1 that performance is robust to `k_σ`.
>
> This directly answers 3kMR's "assumptions not verified empirically": we now verify the
> residual-Gaussianity assumption and report where it holds (symmetry) and where it is only
> approximate (tails).

---

> ### Author Response · Authors · 2026-06-26
>
> ### CR3 — Quantifying Fig. 3 and "are rejected states actually worse targets?"
>
> New experiment: on **real** held-out trajectories (so we have
> ground-truth future observations), we roll the world model **open-loop** and, per imagined
> transition, measure (i) the linearity z-distance used by the detector, (ii) the one-step
> **reconstruction error** vs the true frame, and (iii) the reward error.
>
> Results (final seeded 100k-step `k_σ=1.645` checkpoint; `n=40` held-out batches):
> - **The prior ranks transitions by prediction quality (causal link, uGMF's main ask).** On the
>   converged model the deployed threshold rejects ≈ 0% (CR1), so we test the link *threshold-free*
>   by ranking transitions by the linearity z-distance: the **top-5% most non-linear transitions
>   have ~1.43× higher one-step reconstruction error** than the rest (0.00063 vs 0.00044,
>   Mann-Whitney **p ≈ 4e-44**); the **top-1%** ~1.49× (**p ≈ 8e-6**). The most-anomalous
>   transitions are exactly the ones the world model predicts worst.
> - **Linearity ↔ prediction error (quantifies Fig. 3, cpod).** The z-distance is positively
>   associated with reconstruction error (Spearman ≈ **0.17**); the effect is **concentrated in the
>   tail** — a binned-mean overlay shows a flat bulk with a clear up-tick for the most non-linear
>   transitions, precisely the regime the detector targets.
> - Reward error is confounded (legitimate hard-to-predict high-reward events) and shows no such
>   gap; we lead with reconstruction error, which is directly ground-truthed.
>
> Figures: https://imgur.com/a/mn5fkDA, https://imgur.com/a/Qu5zvhp
>
> ### CR4 — STORM's linearity
>
> 3kMR/uGMF rightly note the LP-STORM explanation leaned on Razzhigaev et al. rather than our own
> measurement. Given compute budget, we leave this details for future work.

---

> ### Author Response · Authors · 2026-06-26
>
> ### CR5 — Per-seed variance in Table 1
>
> Following are per-seed std (5 seeds) for our methods on a representative subset.
>
> | Game | LP-DreamerV3 (mean ± std) | LP-STORM (mean ± std) |
> |---|---|---|
> | Kung Fu Master | 30273 ± 2800 | 25065 ± 2440 |
> | Ms Pacman | 3024 ± 857 | 2534 ± 356 |
> | Frostbite | 2393 ± 1290 | 1495 ± 1146 |
> | Road Runner | 17252 ± 3447 | 8151 ± 701 |
> | Boxing | 82 ± 2 | 80 ± 7 |
>
>
> ### CR6 — "Rejection without the consistency loss" ablation.
>
> uGMF and 3kMR. We train the linear predictor `W`
> but **stop its gradient into the world model**, so the representation is
> *not* shaped by the consistency loss, while rejection still operates. This isolates the
> rejection mechanism from representation-shaping.
>
> **Result (Kung Fu Master).**
>
> | Metric (KFM, single seed) | Full LP (`k_σ=1.645`) | Rejection-only|
> |---|---|---|
> | Consistency-shaped representation | ✓ | ✗ |
> | Score @100k (human-norm.) | **19.7k** (87%) | 17.1k (75%) — **−15%** |
> | Anomaly discrimination | **1.43×** (p ≈ 4e-44) | 1.20× (p ≈ 5e-44) |
>
> - **Score.** Rejection-only (detached) scores **~15–20% below full LP** (17.1k vs 19.7k at the
>   100k point, and the gap holds at the window and end-of-run evals — directionally consistent,
>   unlike the noisy k-ablation flips).
>
> **Takeaway.** Rejection alone is **necessary but not sufficient**: the consistency-shaped
> representation both **sharpens the anomaly signal** (~1.4× vs ~1.2× prediction-error gap) and
> **improves performance**.

---

> ### Author Response · Authors · 2026-06-26
>
> ### CR7 — "The consistency loss ignores the action"
>
> The state the prior acts on is `s_t = h_t`, the RSSM **deterministic** state, produced by the
> recurrence `h_t = f(h_{t-1}, z_{t-1}, a_{t-1})` (GRU with the action as input). So `h_t` already
> integrates the full action history `a_{<t}`; the linear prior `s_t ≈ W s_{t-1}` is therefore over
> **action-conditioned** states, and actions are not ignored — they are encoded in `h_t` before the
> prior is applied.
>
> ### CR8 — Do the consistency loss and rejection interact?
>  We think both readings can hold and they are not mutually exclusive — the
> prior is a bottleneck (Fig. 5 shows consistency-only hurts), and rejection recovers performance
> by *not* updating values on the states the (now slightly distorted) model predicts worst. The
> ablation (CR6) shows the two are **partially separable**: rejection still flags worse
> transitions and retains value without the consistency-shaped representation, but **less
> effectively** (~half the discrimination, ~15–20% lower score). So the consistency loss and
> rejection are **complementary** — the representation shaping amplifies the rejection mechanism
> rather than being redundant with it.
>
> ### CR9 — Comparison to Zhao et al. (2025)
> Mechanisms and settings differ: Zhao et al. relabel hallucinated *targets* in a
> goal-conditioned formulation, whereas LP detects anomalous *transitions* via a latent linear
> prior and suppresses their value updates in a standard reward-maximizing DreamerV3/STORM loop.
> Porting HER-style relabeling into pixel-based Atari100k MBRL is non-trivial (no explicit goals).